# Emissions Released by Forest Fuel in the Daxing'an Mountains, China

Heng Zhang [1,2], Hui Li [1,2], Xinyuan Liu [1,3], Yunjia Ma [1,3], Qing Zhou [1,2], Rula Sa [1,*] and Qiuliang Zhang [2,*]

1   College of Forestry, Inner Mongolia Agricultural University, Hohhot 010019, China; zhangheng_nefu@126.com (H.Z.); lihui47735420@163.com (H.L.); lxyfire@163.com (X.L.); mayunjia0427@163.com (Y.M.); wilsonzhou1224@163.com (Q.Z.)
2   Forest Ecosystem National Observation and Research Station of Daxing'an Mountains, Genhe 022350, China
3   College of Forestry, Northeast Forestry University, Harbin 150040, China
*   Correspondence: sarula213@imau.edu.cn (R.S.); 18686028468@163.com (Q.Z.)

**Abstract:** The large amounts of emissions released by forest fires have a significant impact on the atmospheric environment, ecosystems, and human health. Revealing the main components of emissions released by forest fuel under different combustion states is of great importance to evaluate the impact of forest fires on the ecological environment. Here, a self-designed biomass combustion system was used to simulate the combustion of different parts (i.e., branch, trunk, and bark) of five tree species and branches, and three layers of surface dead fuel (i.e., litter layer, semi-humus layer, and humus layer) of three shrub species, in the Daxing'an Mountains, Inner Mongolia. The emission characteristics of the main gas pollutants (i.e., CO, $CO_2$, HC, and $NO_x$) and $PM_{2.5}$ released under the two combustion states of smoldering and flaming, along with the correlation ratio among emission factors, were measured. The results show that the average amounts of emissions released by different trees and the three layers of surface dead fuel from a smoldering state are higher than those from the flaming state, while shrub combustion shows the opposite. The emissions released by trees, shrubs, and surface dead fuel from the flaming state are ordered from high to low as follows: $CO_2$ > CO > HC > $NO_x$ > $PM_{2.5}$; and from the smoldering state as $CO_2$ > CO > HC > $PM_{2.5}$ > $NO_x$, indicating that the primary emissions under both conditions are mainly due to $CO_2$, CO, and HC, while the emissions of $NO_x$ and $PM_{2.5}$ are dependent on the combustion state—flaming promotes the emission of $NO_x$, while smoldering promotes the emission of $PM_{2.5}$. The average emissions of $PM_{2.5}$ from the branches, bark, and trunks of *Quercus mongolica* are significantly higher than those of the other four tree species in the smoldering state, and the emissions of $PM_{2.5}$ from the five tree species are ordered as follows: bark > branch > trunk. This study will help to further understand the impact of forest fires on the atmospheric environment and ecosystems in Northern China.

**Keywords:** biomass combustion; wood combustion; controlled combustion; incomplete combustion

## 1. Introduction

Biomass combustion releases large amounts of particulate matter, which has a significant impact on the atmospheric environment, ecosystem, and human health [1–4]. Forest ground-cover fuel is the basic material of forest combustion and a primary cause of forest fires [5]. Forest fires have duality: on the one hand, they can maintain the balance and stability of forest ecosystems; on the other hand, they can lead to the destruction of the forest's ecological balance [6]. Wildfires and prescribed fires (preceded by harvest or not) can serve to promote giant sequoia regeneration, providing that the fire intensity is sufficient to create canopy gaps, increase understory light, and remove surface litter [7]. In boreal North America, black spruce shapes forest flammability, and depends on fire for regeneration [8]. Transition of forests to savannah suggests that fire disturbance can be a major driver of biome change [9]. The emissions released by the combustion of forest fuel greatly increase the contents of greenhouse gases in the atmosphere, affecting many atmospheric chemical

processes, and constituting a significant natural disturbance factor that accelerates global warming [10]. The presence of open flame—i.e., flaming combustion—has a significant impact on the chemical composition of emissions and plume dynamics [11]. Smoldering combustion is the driver of wildfires in peat lands, such as those that cause episodes of haze in Southeast Asia, North America, and Northeast Europe [12]. Gas emissions from smoldering fires differ significantly from those from flaming fires. First, the emission rate per unit of area is much lower, and has different chemistry. Smoldering is characteristically an incomplete form of combustion, which releases species and quantities that substantially differ from those in stoichiometric and complete combustion [12]. When external conditions such as temperature, wind speed, and wind direction change, smoldering may change to flaming.

In the year 2000 alone, the worldwide area of forest resources lost due to forest fires in the world was $3.36$–$3.50 \times 10^6$ ha, in which the mass of biomass burned was more than 2814 Gt [13,14]. The influence of various factors, such as temperature and humidity, can lead to a diverse range of flaming and smoldering phenomena during the combustion process of forest stands and their biomass [15]. When a surface fire burns the ground cover and shrub understory of a forest, it causes damage to the branches, bark, and trunks of trees [16,17] through a combustion process that includes 20% smoldering and 80% flaming [18], which can lead to wildfires that endanger the lives of humans and other organisms [19].

The impact of emissions released from forest fires on global climate change is a critical research topic. For example, Li et al. [20] and Liu et al. [21] discussed the impact of air pollutants from forest fires on the atmospheric environment. Guo [22] calculated the biomass using the volume–biomass inversion model, estimated the carbon emissions of forest fires in the Sanming area based on the biomass information, and quantitatively analyzed the impact of forest fires on the ecosystem's carbon cycle. Kasischke et al. [23] estimated a range of carbon emissions based on different assumptions of the depth of burning, and concluded that an increase in northern fires can impact atmospheric $CO_2$ in the Northern Hemisphere. Rogers et al. [24] used remote sensing imagery, climate reanalysis data, and forest inventories to evaluate differences in boreal fire dynamics between North America and Eurasia, along with their key drivers, and concluded that species-level traits must be considered in global evaluations of the effects of fires on emissions and climate. Kondo et al. [25] related the combustion phase of the fire as represented by the modified combustion efficiency (MCE) to the emission ratios between black carbon (BC) and other species, and concluded that the difference in the BC/CO emission ratios is likely due to the difference in MCE. Previously, Levine [26] analyzed the abundant forest fires of Indonesia in 1997 in terms of their combustion area, combustion efficiency, combustion biomass, biomass load, and the emission ratio of each component. Finally, Okoshi et al. [27] found that the concentrations of fine particulate matter in small areas and low-quality forest fires were higher than those in large areas and high-quality forest fires.

The Daxing'an Mountains of Inner Mongolia are a key fire risk area in China [28]. In May 2017, a huge forest fire occurred in the north of the Bilahe Forestry Bureau in the Daxing'an Mountains. The burned area was 11,500 hm², 60% of which was forest land, and the total affected forest area was 8281.58 ha. A total of 9430 people (including 3290 people from the armed police forest force) and 14 aircraft of various types were used to fight the fire. The Daxing'an Mountains forest area in Inner Mongolia is an important part of the boreal forest belt of Eurasia, with a high forest coverage rate of 77.99% [17] and a high fire frequency.

The growing use of woody biomass in the energy sector for renewable energy is considered to show potential, as it is a stable and predictable source providing a number of positive functions for electrical power systems, in addition to the energy, economic, and environmental benefits [29]. Here, we simulate the combustion of different trees, shrubs, and surface dead fuel in the Daxing'an Mountains, Inner Mongolia, along with the emission characteristics of forest fire emissions composed of CO, HC, $CO_2$, $NO_x$, and $PM_{2.5}$, and

the correlation ratios between various emission factors under the two combustion states of smoldering and flaming, to estimate fire emissions at the local level.

## 2. Materials and Methods

### 2.1. Study Area

The Daxing'an Mountains of the Inner Mongolia Bilahe Forestry Bureau (Figure 1) are located at 122°44′00″–123°55′00″ E and 49°00′40″–49°54′00″ N. The region is influenced by topography and a cold monsoon current, meaning that a cold temperate continental monsoon climate prevails in this area. Here, the annual average temperature is ~−1 °C, the extreme maximum temperature is 35.4 °C, and the extreme minimum temperature is −46.0 °C. The annual average precipitation is 479.4 mm, mainly occurring from June to August. The frost-free period is ~130 days, and the relative humidity is 70%–75%. The change in terrain elevation across the forest area is small, from 377 m to 933 m a.s.l. [30]. The total forest area is 5.33 million ha, the forest coverage is 65.6%, and the total wood volume is 36.21 million m$^3$.

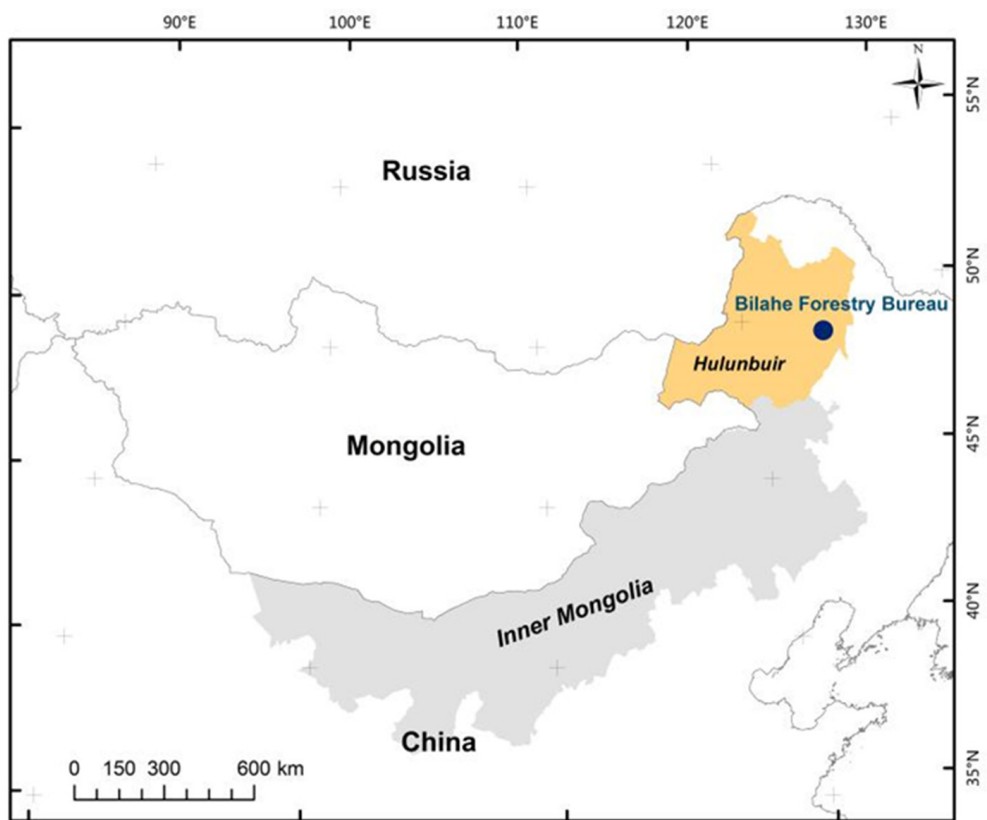

**Figure 1.** Map showing the location of the boreal forest study area in China where the forest fire fuel sources were sampled.

### 2.2. Sampling Plots and Sample Collection

In October 2017, five typical trees—*Quercus mongolica* (MGL), *Betula platyphylla* (BH), *Larix gmelinii* (LYS), *Populus davidiana* (SY), and *Betula dahurica* (HH)—and three typical shrubs—*Corylus heterophylla Fisch* (PZ), *Lespedeza bicolor Turcz* (HZZ), and *Rhododendron dauricum* L (XADJ)—were chosen on a square grid with a fixed side length of 20 m × 20 m set by the method of system distribution. Corresponding components of each tree species (branch, bark, and trunk parts) and shrub species (branches), along with three layers of surface dead fuel (litter, semi-humus, and humus), were then sampled. Since our study does not focus on the impact of biomass moisture content of the pollutant emissions, all samples were naturally air-dried to a constant weight over three days. To ensure that the samples were collected under the same environmental variables, and to reduce the errors

caused by the deposition of particulate matter on the plant leaves by external factors such as air pollution, the sampling sites were located far away from urban areas and highways.

Our methods are based on large-scale emission estimation data; thus, the tree and shrub species did not need to have accurate forest ages. All tree and shrub species were selected from mature individuals with the same slope direction, and the branch, bark, and trunk samples were all 1000 g, which were then placed in different kraft bags. The surface dead fuel samples were divided into three layers: the litter layer (4 cm below the surface), semi-humus layer (2 cm below the litter layer), and humus layer (1.5 cm below the semi-humus layer); all of these also had a sample weight of 1000 g, and were also bagged as described above. Each sample was heated to a constant weight in an oven at 105 °C, and each sample was cut to a length of about 5 cm, each weighing 15 g (accuracy: 0.01 g), to facilitate full combustion, and the processed branch, bark, and trunk samples were divided into 12 groups for combustion tests (six times each for smoldering and for flaming), placed in different kraft bags with good ventilation, labeled, and stored in a cool place.

### 2.3. Combustion Gas Collection

There are differences in the flue gas and particulate matter released by fuel burning under differing states of combustion. Thus, a self-designed biomass combustion device [31] (Figure 2) capable of smoldering and flaming was used here to carry out the indoor simulated combustion experiment. An air pump system kept the air at a sufficient level in the combustion chamber to simulate the combustion that occurs in the open natural environment.

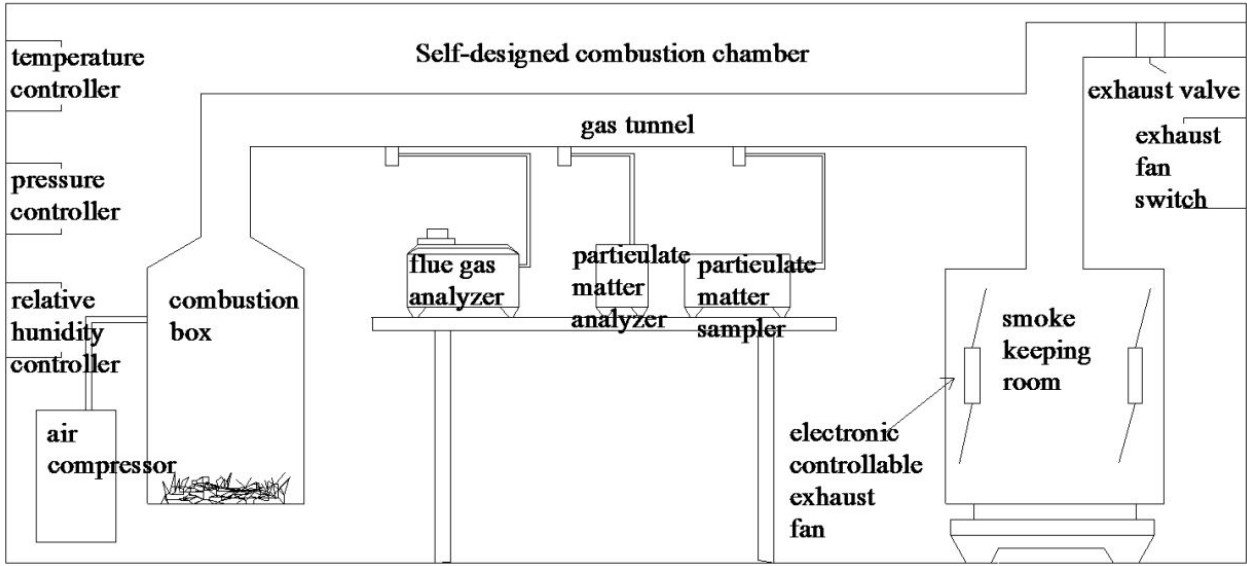

**Figure 2.** Schematic diagram of the biomass combustion device used to simulate smoldering and flaming of various forest fire fuel sources.

During the simulation experiment, a temperature controller and a pressure controller were used to adjust the required testing temperature, so as to simulate the two combustion states of smoldering and flaming, for which the modified combustion efficiency (*MCE*) was used to indicate the combustion states during the simulation experiment [30,32], as follows:

$$MCE = \frac{\Delta C_{CO_2}}{\Delta C_{CO_2} + \Delta C_{CO}} \tag{1}$$

where $\Delta C_{CO_2}$ and $\Delta C_{CO}$ are the changes in ρ ($CO_2$) and ρ (CO) (mg/m$^3$), respectively; when flaming, $MCE \geq 0.99$; when smoldering, $0.65 \leq MCE \leq 0.85$. Through pre-tests of smoldering and flaming, the temperatures under the different combustion states [19,33,34] were measured to be 180 °C and 290 °C, respectively. Before starting the combustion test, we first adjusted the temperature controller to set the temperature of the combustion

chamber to 180 °C or 290 °C to simulate smoldering or flaming states, respectively; we then turned on the flue gas analyzer and the particulate matter analyzer, calibrated them with standard gas, and added 15 g of each processed sample for combustion.

Before adding the samples for combustion, we tested the emissions in the combustion chamber as a control, added the sample to the combustion device to burn, and immediately turned on the particulate matter sampler to collect $PM_{2.5}$. We then used the flue gas analyzer and the particulate matter analyzer to monitor the flue gas in the combustion chamber in real time to display the concentrations of various pollutants in the flue gas, which were used to calculate the emission factors of particulate matter and the MCE.

The collection of particulate matter in both combustion states was continued until no smoke emission remained. An inorganic quartz membrane was used for the collection of particulate matter. Before collection, it was placed in a muffle furnace at 450 °C for 3 h to remove water and volatile substances, placed in a drying dish for 24 h until constant weight, and then weighed and recorded (balance accuracy: $10^{-5}$ g). The membranes were then wrapped in tin foil after sampling and stored in a drying dish for subsequent component analysis tests.

### 2.4. Calculation of Emission Factors

The emission ratio is the ratio of the mass of carbon-containing gas from the forest fire to the total mass of carbon lost in the combustion process [35]. The amounts of carbon and carbon-containing gas released from forest fires can be measured by the emission ratio method or the emission factor method; however, the latter approach is considered more reliable [36]. We therefore applied this method here, based on the carbon conservation principle [37]. The basic premise of this method is that if the combustion reaction of the sample is sufficient, all carbon present will be in the form of gaseous $CO_2$, CO, total hydrocarbons, and particulate matter. The emission factors of CO, HC, $CO_2$, $NO_x$, and $PM_{2.5}$ were then calculated as follows: First, we defined an incomplete combustion coefficient (*PIC*):

$$PIC = \frac{C_{C-CO} + C_{C-PM} + C_{C-THC}}{C_{C-CO_2}} \quad (2)$$

where $C_{C-CO}$, $C_{C-PM}$, $C_{C-THC}$, and $C_{C-CO_2}$ are the carbon emissions of CO, PM, THC, and $CO_2$ (g), respectively [11,26–29]. Next, we calculated the emission factor of $CO_2$:

$$EF_{CO_2} = \frac{(C_f - C_a) \times f_{CO_2}}{(PIC + 1) \times M} \quad (3)$$

where $EF_{CO_2}$ is the emission factor of $CO_2$ (g/kg), $C_f$ is the mass of fuel carbon (g), $C_a$ is the mass of ash carbon (g), $f_{CO_2}$ is the medium carbon of CO and the conversion factor of $CO_2$ (i.e., 44/12 = 3.67), and $M$ is the mass of fuel (kg) [11,26–29]. Finally, we calculated $EF_i$:

$$EF_i = \frac{C_i}{C_{CO_2} \times EF_{CO_2}} \quad (4)$$

where $EF_i$ is the emission factor of the target compound (g/kg), $C_i$ is the mass concentration of the target compound (mg/m$^3$), $C_{CO_2}$ is the $CO_2$ concentration (mg/m$^3$), and $EF_{CO_2}$ is the $CO_2$ emission factor (mg/m$^3$) [19,30,32–34].

### 2.5. Statistical Analysis

Microsoft Excel 2018 software was used to collect basic data and establish a database; ArcGIS software was used to draw distribution maps of the study area; SPSS 22.0 software was used to perform relevant statistical analysis; and Origin 2018 software was used to draw charts.

## 3. Results

*3.1. Emissions Released by Forest Fuel Combustion*

3.1.1. Emissions Released by Trees under Different Combustion States

Emission factors are the basis for the quantitative study of pollutant emissions, and combustion states can significantly affect pollutant emission factors. The emission characteristics of CO, HC, $CO_2$, $NO_x$, and $PM_{2.5}$, as well as the correlation ratios between the pollutant emission factors of the five tree species, were determined under two simulated combustion states.

There were significant differences in pollutant emissions released by the combustion of different parts of trees under different combustion states (Figure 3). The overall differences in pollutant emissions released by the combustion of trees under different combustion states, ordered from greatest to least, was trunk > branch > bark. The main emission factors were CO, HC, and $CO_2$, whereas $NO_x$ and $PM_{2.5}$ were relatively minor. There were significant differences in the emission factors of BH between the two combustion states, while there were significant differences in the CO, $CO_2$, and $NO_x$ of LYS. Figure 3 shows the amounts of emissions released by five tree species in the two combustion states. The results show that $CO_2$ had the largest values, in the range of 2415.04–4828.58 g/kg, followed by CO and HC, in the range of 632.67–1245.64 g/kg and 111.38–962.44 g/kg, respectively. $NO_x$ and $PM_{2.5}$ were emitted in minimal amounts, in the range of 1.94–19.39 g/kg and 4.68–30.29 g/kg, respectively. When smoldering, SY had the highest emissions of $CO_2$; BH had significantly higher emissions of HC than the other tree species; MGL had the highest emissions of CO, but with little difference between the five tree species, while its emissions of $PM_{2.5}$ were significantly higher than those of the other tree species. LYS had the highest emissions of $NO_x$. When flaming, except for MGL—which had significantly higher emissions of $NO_x$ than the other tree species—this pollutant was emitted in similar amounts among all tree species. Of the five emission factors, the emission characteristics of $PM_{2.5}$ were the most significantly different, being significantly higher for each tree species when smoldering than flaming, with corresponding $PM_{2.5}$ emissions for MGL, BH, LYS, HH, and SY when smoldering of 6.1, 1.3, 1.6, 1.8, and 2.5 times those when flaming, respectively. The mass of other emission factors of the different tree species increased and decreased in the two combustion states. The total mass of pollutants emitted from the tree species' combustion, ordered from high to low, was as follows: BH > HH > SY > LYS > MGL.

The different amounts of emissions released by forest fuel combustion under different combustion states can be expressed as follows:

$$Change\ of\ emission\ factor(n) = Smoldering/Flaming \tag{5}$$

The change in tree CO was 1.14, in HC was 1.76, in $CO_2$ was 0.82, in $NO_x$ was 0.80, and in $PM_{2.5}$ was 2.59. The increase in $PM_{2.5}$ was the largest, with $n > 1$, indicating that its increase was more significant under smoldering conditions, mainly driven by MGL ($nPM_{2.5}$ = 6.11). The largest decrease was for $NO_x$ with $n < 1$, indicating that its decline was more significant under flaming conditions, and was largely due to MGL ($nNO_x$ = 0.36).

There were significant differences in the pollutant emissions in the process of tree combustion between the two combustion states (Table 1). The differences in carbon-containing gas emissions from the combustion of different parts of the same tree species were related to the carbon content, physicochemical properties, and combustion efficiency of the bark, trunks, and branches. Specifically, $NO_x$ released from parts of SY, HH, and LYS was significantly different between combustion states, while CO from parts of BH and LYS was significantly different. When smoldering, the highest emission of CO was from BH bark, and when flaming, the highest was from LYS bark. CO was mainly concentrated in bark, and the highest and lowest HC emissions (smoldering and flaming, respectively) were released from BH branches. These results show that different combustion states have a particularly important effect on the HC emissions released from BH branches.

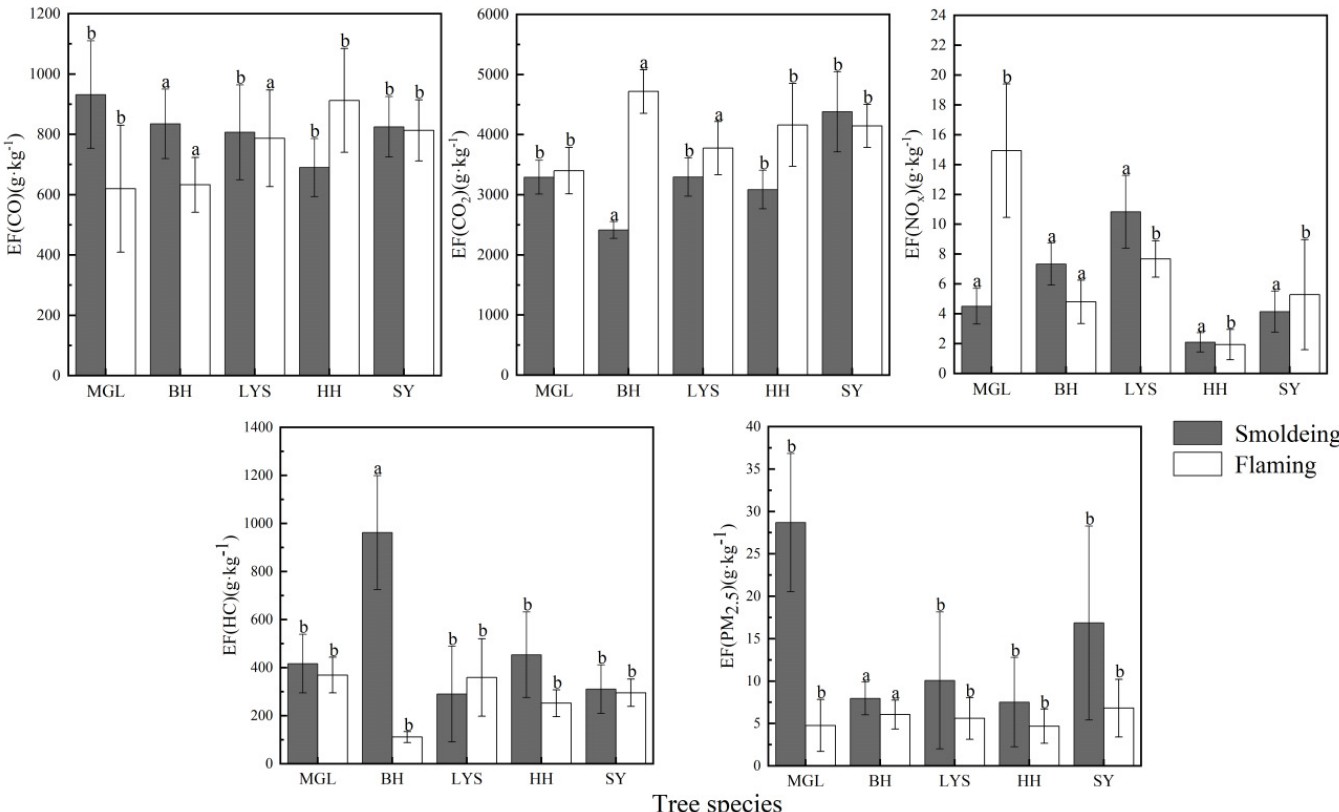

**Figure 3.** Emissions released by five trees under different combustion states. Note: SY, *Populus davidiana*; BH, *Betula dahurica*; LYS, *Larix gmelinii*; HH, *Betula platyphylla*; MGL, *Quercus mongolica*. Identical letters indicate no significant difference, while different letters indicate a significant difference between the treatment and the control; the same below.

### 3.1.2. Emissions Released by Shrubs under Different Combustion States

The emissions of CO, $CO_2$, and HC from shrubs under smoldering conditions were ordered as PZ > XADJ > HZZ; while those of $PM_{2.5}$ were XADJ > HZZ > PZ. When flaming, the emissions of $CO_2$ and $NO_x$ were ordered as HZZ > PZ > XADJ, while those of CO and $PM_{2.5}$ were XADJ > PZ > XADJ, and those of HC were PZ > HZZ > XADJ.

These results indicate that the emissions released by different shrub species under different combustion states are different, owing to the structural components, physiochemical properties, initial carbon content, and other factors of different shrub species.

The change in shrub CO was 1.00, in $CO_2$ was 1.03, in $NO_x$ was 1.14, in HC was 0.92, and in $PM_{2.5}$ was 4.83. The change in $PM_{2.5}$ was the largest, with $n > 1$, indicating that its increase was more significant when smoldering, mainly driven by HZZ (n$PM_{2.5}$ = 6.88). The change in HC was small, with $n < 1$, indicating that its emissions were lower under smoldering than that under flaming, but the decrease in HC was the clearest (nHC = 0.75). These results indicate that different combustion states have a significant effect on the emissions of different emission factors of HCs.

### 3.1.3. Emissions Released by Surface Dead Fuel under Different Combustion States

For surface dead fuel, the emissions of HH were the highest, while those of LYS were the lowest; their emission factors were mainly composed of CO, HC, and $CO_2$, whereas $NO_x$ and $PM_{2.5}$ contributed relatively little (Figure 4), consistent with the results of Li [38]. There were significant differences in CO, HC, and $NO_x$ emissions from MGL between the two combustion states, as well as for $CO_2$, CO, and $PM_{2.5}$ emissions from HH.

**Table 1.** Emissions of tree parts under different combustion states (mean ± standard deviation; g/kg).

| Tree Parts | | Smoldering | | | | |
|---|---|---|---|---|---|---|
| | | CO | $CO_2$ | $NO_x$ | HC | $PM_{2.5}$ |
| MGL | Trunk | 283.61 ± 35.85 b | 968.75 ± 90.43 b | 1.78 ± 0.29 b | 76.38 ± 9.56 b | 6.54 ± 2.98 b |
| | Bark | 290.90 ± 37.95 b | 1149.33 ± 52.36 b | 1.63 ± 0.56 b | 100.86 ± 26.47 b | 15.61 ± 4.00 b |
| | Branch | 357.12 ± 104.51 b | 1174.51 ± 140.02 b | 1.11 ± 0.35 b | 239.37 ± 85.70 b | 6.56 ± 1.17 b |
| BH | Trunk | 312.63 ± 13.98 a | 817.11 ± 45.55 b | 0.77 ± 0.16 b | 354.21 ± 53.02 b | 1.20 ± 0.19 b |
| | Bark | 379.53 ± 51.32 a | 1012.87 ± 13.1 b | 5.91 ± 0.80 b | 75.33 ± 11.84 b | 6.36 ± 1.25 b |
| | Branch | 142.39 ± 49.99 a | 584.06 ± 80.5 b | 0.64 ± 0.45 b | 532.90 ± 171.95 b | 0.40 ± 0.50 b |
| LYS | Trunk | 228.86 ± 92.87 b | 1200.29 ± 165.3 b | 1.63 ± 0.93 a | 88.70 ± 80.22 b | 1.84 ± 1.17 b |
| | Bark | 362.18 ± 46.89 b | 944.65 ± 67.68 b | 6.65 ± 0.42 a | 92.33 ± 34.95 b | 4.89 ± 3.72 b |
| | Branch | 215.74 ± 17.53 b | 1148.98 ± 85.5 b | 2.56 ± 1.08 a | 109.57 ± 83.84 b | 3.36 ± 3.19 b |
| HH | Trunk | 241.19 ± 22.47 b | 942.4 ± 93.59 b | 0.30 ± 0.16 b | 185.94 ± 80.25 b | 2.07 ± 1.60 b |
| | Bark | 247.72 ± 4.79 b | 983.13 ± 74.65 b | 0.20 ± 0.06 b | 142.47 ± 52.90 b | 1.88 ± 0.56 b |
| | Branch | 201.08 ± 69.02 b | 1160.53 ± 151.68 b | 1.59 ± 0.42 b | 125.14 ± 44.98 b | 3.57 ± 3.09 b |
| SY | Trunk | 279.88 ± 47.79 b | 1454.46 ± 359.87 b | 0.77 ± 0.51 a | 92.85 ± 46.45 b | 7.40 ± 6.52 b |
| | Bark | 228.72 ± 33.9 b | 1287.40 ± 205.08 b | 0.38 ± 0.10 a | 120.76 ± 29.70 b | 4.73 ± 3.54 b |
| | Branch | 316.43 ± 18.28 b | 1638.28 ± 100.1 b | 2.99 ± 0.77 a | 97.18 ± 24.47 b | 4.74 ± 1.36 b |
| Tree Parts | | Flaming | | | | |
| | | CO | $CO_2$ | $NO_x$ | HC | $PM_{2.5}$ |
| MGL | Trunk | 195.62 ± 81.01 b | 1088.14 ± 168.06 b | 8.53 ± 3.40 b | 153.66 ± 36.47 b | 1.59 ± 1.35 b |
| | Bark | 211.74 ± 75.17 b | 1157.36 ± 113.72 b | 2.25 ± 0.34 b | 110.69 ± 19.93 b | 1.59 ± 1.53 b |
| | Branch | 212.10 ± 54.34 b | 1156.06 ± 104.56 b | 4.15 ± 0.72 b | 104.65 ± 17.91 b | 0.29 ± 0.18 b |
| BH | Trunk | 149.45 ± 18.17 a | 1642.46 ± 85.86 b | 1.21 ± 0.18 b | 34.91 ± 3.88 b | 2.35 ± 0.22 b |
| | Bark | 164.52 ± 59.22 a | 1696.38 ± 182.01 b | 2.89 ± 0.69 b | 46.03 ± 5.15 b | 3.40 ± 1.23 b |
| | Branch | 318.71 ± 13.35 a | 1379.26 ± 95.94 b | 0.70 ± 0.60 b | 30.44 ± 13.57 b | 0.30 ± 0.24 b |
| LYS | Trunk | 221.01 ± 28.39 a | 1669.54 ± 80.09 a | 0.68 ± 0.10 a | 67.00 ± 17.11 b | 2.31 ± 0.94 b |
| | Bark | 411.39 ± 78.71 a | 910.44 ± 257.14 a | 6.16 ± 0.66 a | 187.82 ± 123.11 b | 2.05 ± 0.78 b |
| | Branch | 154.50 ± 53.02 a | 1195.89 ± 106.24 a | 0.83 ± 0.45 a | 104.36 ± 20.71 b | 1.24 ± 0.75 b |
| HH | Trunk | 275.73 ± 110.6 b | 1615.11 ± 543.01 b | 0.11 ± 0.05 b | 85.87 ± 30.75 b | 1.28 ± 0.79 b |
| | Bark | 381.32 ± 44.54 b | 1493.40 ± 117.46 b | 0.82 ± 0.40 b | 57.85 ± 14.86 b | 1.48 ± 0.45 b |
| | Branch | 255.49 ± 17.16 b | 1053.47 ± 27.9 b | 1.01 ± 0.58 b | 108.30 ± 10.09 b | 1.91 ± 0.78 b |
| SY | Trunk | 290.29 ± 49.19 b | 1415.63 ± 163.41 b | 0.39 ± 0.36 b | 61.52 ± 19.49 a | 1.26 ± 0.51 b |
| | Bark | 251.03 ± 10.63 b | 1307.44 ± 154.32 b | 3.02 ± 3.24 b | 161.01 ± 36.75 a | 1.00 ± 0.26 b |
| | Branch | 271.96 ± 41.29 b | 1422.70 ± 39.41 b | 1.87 ± 0.99 b | 72.86 ± 0.61 a | 4.57 ± 2.63 b |

Note: Different lowercase letters in the same column indicate significant difference $p < 0.05$), the same below.

The change in surface dead fuel CO emissions was 1.10, in $CO_2$ was 0.82, in $NO_x$ was 0.57, in HC was 1.20, and un $PM_{2.5}$ was 1.02, being the largest for HC, with $n > 1$, indicating its prevalence via smoldering, which was mainly influenced by BH (nHC = 2.72). An opposite change in $NO_x$ was observed, with $n < 1$, indicating that its emissions were lessened during smoldering, decreasing the most in SY (n$NO_x$ = 0.29). When the combustible volatiles from the surface dead fuel combustion process reach a critical concentration, the fuel enters the flaming stage. At that time, the emission amount of CO decreases, while that of $CO_2$ increases rapidly. When the influencing factors, such as combustible volatiles or oxygen concentration, are lower than a critical limit, the flaming stage shifts into the smoldering stage, and the emissions of $CO_2$ decrease, while those of CO begin to increase [39].

Surface dead fuel was divided into three layers: litter, semi-humus, and humus. The $NO_x$, $CO_2$, and HC in each layer of SY differed significantly between the two combustion states, with significant differences in CO contents in each layer of both BH and HH (Table 2). The emissions of CO from HH litter were highest when flaming and lowest when smoldering, indicating that they were significantly affected by different combustion states. Conversely, the emissions of $CO_2$ from the SY humus layer were lowest when flaming and highest when smoldering, also indicating that they were significantly affected by different

combustion states. $CO_2$ was released primarily from the surface dead fuel humus layer, and the main emission factor of surface dead fuel was generally $CO_2$.

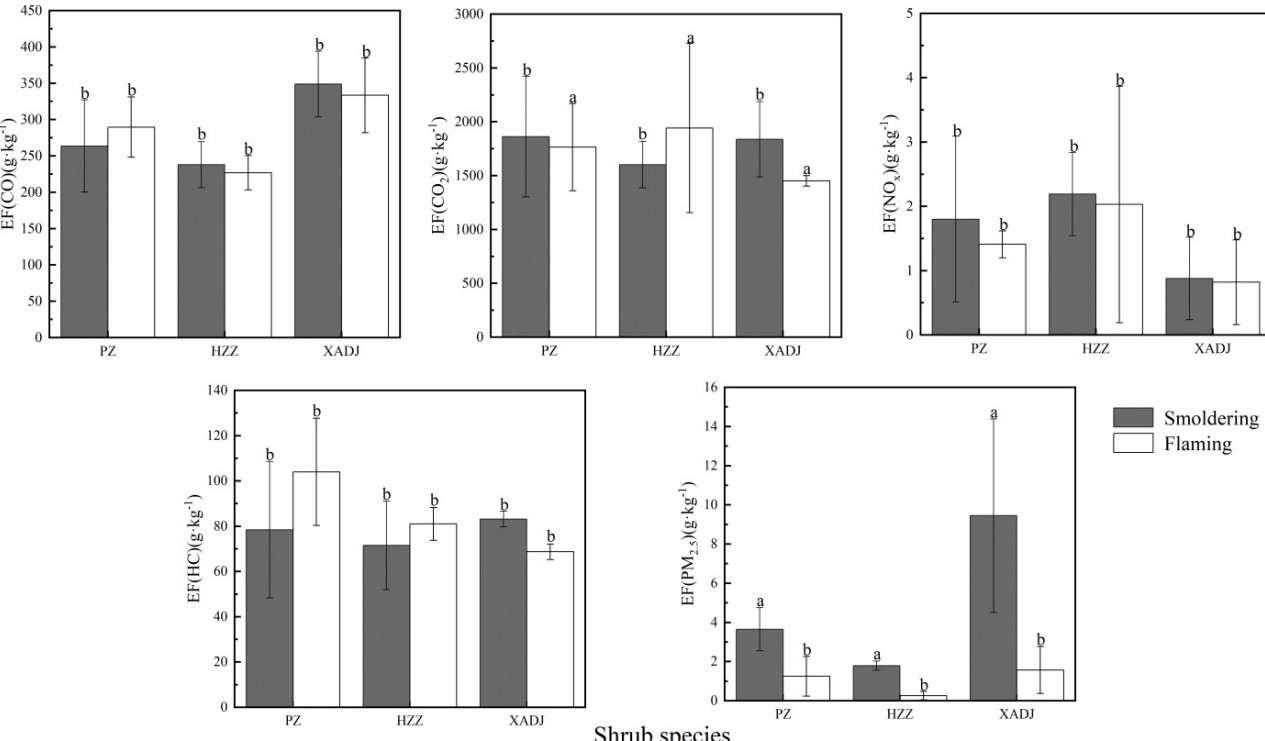

**Figure 4.** Emissions released by three shrub species under different combustion states. Note: XADJ, *Rhododendron dauricum* L.; HZZ, *Lespedeza bicolor Turcz*; PZ, *Corylus heterophylla Fisch*; Identical letters indicate no significant difference, while different letters indicate a significant difference between the treatment and the control;the same below.

**Table 2.** Emissions of different surface dead fuel layers under different combustion states (mean $\pm$ standard deviation; g/kg).

| Surface Dead Fuel | | Flaming | | | | |
|---|---|---|---|---|---|---|
| | | CO | CO₂ | NOₓ | HC | PM₂.₅ |
| MGL | Litter | 175.60 ± 28.84 b | 1505.94 ± 176.09 b | 3.45 ± 0.66 b | 61.59 ± 24.07 b | 2.84 ± 0.92 b |
| | Semi-humus | 172.75 ± 17.83 b | 1853.95 ± 639.83 b | 3.79 ± 0.49 b | 44.89 ± 31.28 b | 1.77 ± 0.84 b |
| | Humus | 193.20 ± 15.04 b | 1561.70 ± 99.09 b | 3.00 ± 0.72 b | 32.60 ± 21.52 b | 2.94 ± 0.36 b |
| BH | Litter | 128.25 ± 10.90 b | 1897.00 ± 414.18 b | 3.86 ± 1.05 b | 34.14 ± 5.04 b | 2.57 ± 0.51 b |
| | Semi-humus | 179.02 ± 87.65 b | 1631.16 ± 95.20 b | 2.43 ± 0.30 b | 38.37 ± 5.56 b | 2.52 ± 0.47 b |
| | Humus | 249.78 ± 101.04 b | 1683.31 ± 288.86 b | 1.74 ± 0.62 b | 37.58 ± 10.54 b | 3.09 ± 0.73 b |
| LYS | Litter | 154.38 ± 81.88 b | 1373.26 ± 98.98 b | 2.90 ± 0.58 b | 39.49 ± 5.58 a | 1.14 ± 0.76 b |
| | Semi-humus | 169.42 ± 117.28 b | 1487.65 ± 152.77 b | 3.22 ± 0.24 b | 19.15 ± 11.39 a | 2.77 ± 1.05 b |
| | Humus | 154.50 ± 53.02 b | 1647.46 ± 289.74 b | 3.44 ± 1.13 b | 115.90 ± 15.13 a | 3.80 ± 0.64 b |
| HH | Litter | 298.38 ± 28.08 a | 1698.27 ± 174.98 b | 2.42 ± 0.53 b | 131.23 ± 39.56 b | 4.21 ± 037 b |
| | Semi-humus | 273.93 ± 64.81 a | 1627.33 ± 238.78 b | 3.07 ± 0.78 b | 140.54 ± 4.36 b | 1.85 ± 0.64 b |
| | Humus | 589.46 ± 42.89 a | 3965.67 ± 624.00 b | 2.55 ± 0.60 b | 125.62 ± 11.71 b | 3.62 ± 0.81 b |
| SY | Litter | 237.98 ± 82.63 b | 1579.08 ± 79.15 b | 4.13 ± 1.21 b | 100.97 ± 3.87 a | 2.71 ± 0.74 b |
| | Semi-humus | 227.82 ± 25.29 b | 1313.04 ± 61.24 b | 11.09 ± 1.58 b | 129.12 ± 2.63 a | 2.55 ± 0.85 b |
| | Humus | 161.21 ± 18.83 b | 1284.89 ± 18.54 b | 9.05 ± 1.47 b | 123.01 ± 9.57 a | 2.80 ± 0.44 b |

**Table 2.** *Cont.*

| Surface Dead Fuel | | Smoldering | | | | |
|---|---|---|---|---|---|---|
| | | CO | CO$_2$ | NO$_x$ | HC | PM$_{2.5}$ |
| MGL | Litter | 192.21 ± 48.53 b | 1096.51 ± 48.07 b | 2.89 ± 0.34 a | 60.25 ± 12.42 b | 3.80 ± 0.10 b |
| | Semi-humus | 217.20 ± 25.74 b | 1246.24 ± 91.17 b | 1.29 ± 0.47 a | 70.57 ± 18.63 b | 2.82 ± 0.30 b |
| | Humus | 222.80 ± 23.64 b | 1161.09 ± 22.85 b | 2.87 ± 0.30 a | 95.31 ± 27.43 b | 3.36 ± 0.50 b |
| BH | Litter | 223.00 ± 76.48 a | 1364.50 ± 142.79 b | 1.05 ± 0.34 b | 106.88 ± 27.46 b | 3.75 ± 0.47 b |
| | Semi-humus | 372.44 ± 8.49 a | 1336.58 ± 162.85 b | 2.29 ± 0.42 b | 80.80 ± 3.23 b | 3.34 ± 0.37 b |
| | Humus | 481.87 ± 57.72 a | 1403.48 ± 154.86 b | 1.41 ± 0.40 b | 111.63 ± 20.84 b | 3.15 ± 0.16 b |
| LYS | Litter | 220.71 ± 69.31 b | 1158.28 ± 129.28 b | 1.28 ± 0.52 b | 101.61 ± 23.81 a | 3.77 ± 0.26 b |
| | Semi-humus | 273.26 ± 14.51 b | 1316.19 ± 54.10 b | 2.46 ± 0.61 b | 19.93 ± 5.55 a | 3.67 ± 0.72 b |
| | Humus | 217.28 ± 19.04 b | 1126.49 ± 64.62 b | 2.72 ± 0.43 b | 86.45 ± 6.22 a | 2.18 ± 0.78 b |
| HH | Litter | 164.10 ± 63.37 b | 1319.40 ± 240.1 b | 1.98 ± 0.46 b | 113.59 ± 14.81 b | 1.77 ± 0.51 b |
| | Semi-humus | 221.20 ± 26.04 b | 1393.88 ± 106.93 b | 4.00 ± 1.41 b | 93.1 ± 4.33 b | 1.42 ± 0.28 b |
| | Humus | 213.03 ± 76.03 b | 1843.35 ± 342.99 b | 2.75 ± 0.34 b | 108.59 ± 6.30 b | 1.81 ± 0.28 b |
| SY | Litter | 206.95 ± 58.04 b | 1576.82 ± 146.37 b | 1.45 ± 0.46 b | 139.80 ± 6.52 b | 2.14 ± 0.55 b |
| | Semi-humus | 270.71 ± 30.03 b | 1835.19 ± 217.23 b | 2.80 ± 0.98 b | 129.22 ± 21.10 b | 3.22 ± 1.17 b |
| | Humus | 264.21 ± 37.00 b | 2017.39 ± 104.27 b | 2.77 ± 0.49 b | 93.84 ± 10.36 b | 2.62 ± 0.84 b |

Note: Different lowercase letters in the same column indicate significant difference *p* < 0.05), the same below.

The emissions of NO$_x$ were highest from the SY semi-humus layer when flaming, but were highest from the HH semi-humus layer when smoldering. NO$_x$ emissions were mainly released from the semi-humus layer. Whether smoldering or flaming, HC was lowest in the LYS semi-humus layer; hence, the latter was negligibly affected by different combustion states. The emissions of PM$_{2.5}$ from the LYS litter layer were lowest when flaming, but became the highest when smoldering (Figure 5).

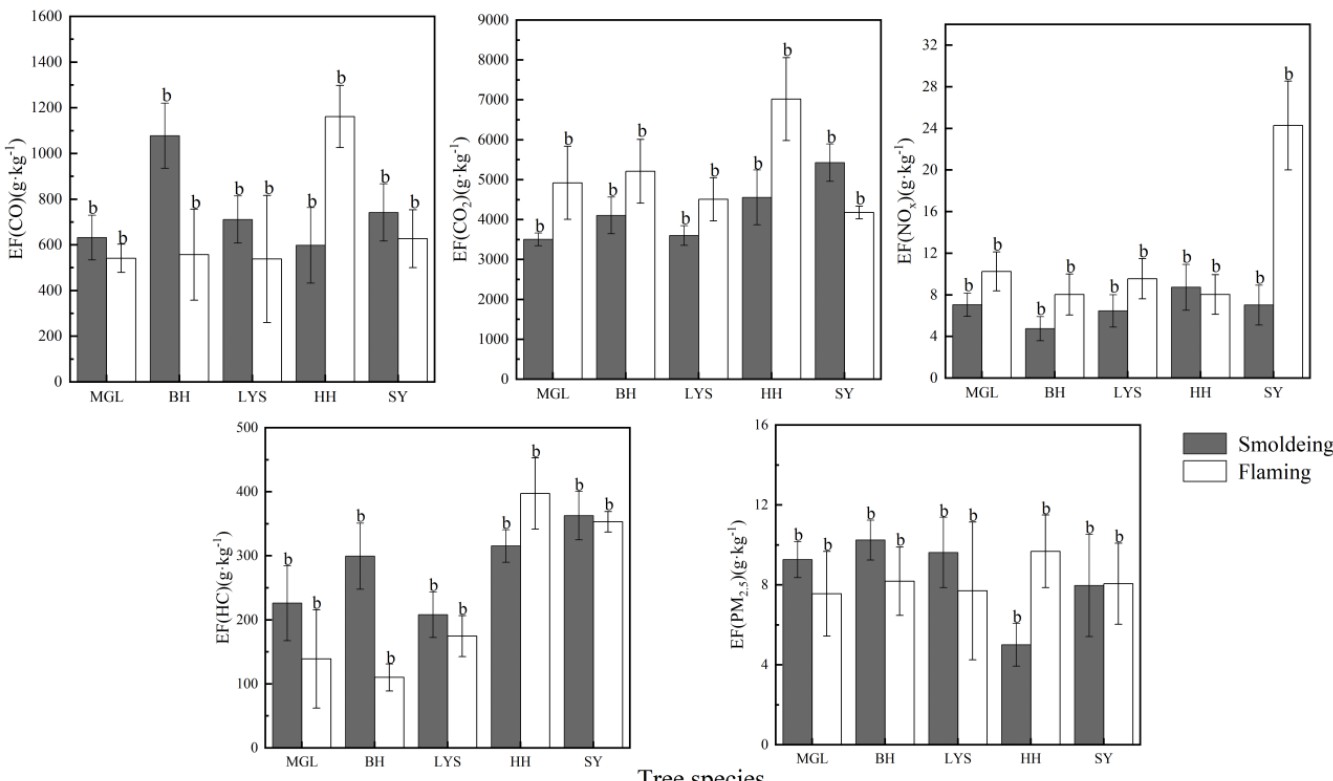

**Figure 5.** Emissions released by different layers of five tree species under different combustion states. Identical letters indicate no significant difference, while different letters indicate a significant difference between the treatment and the control.

## 4. Discussion

Emission factors are an important basis for pollutant emission estimation and environmental risk assessment, and are an important part of the emission characteristics of pollutants from forest biomass combustion [39–41]. Under similar experimental conditions, due to the different lignin contents among different tree species and parts [42], the total amounts of pollutants released by the combustion of different trees, shrubs, and surface dead fuel—as well as their different parts—were significantly different in the Daxing'an Mountains of Inner Mongolia. When smoldering, the highest emissions of CO were 379.53 g/kg from BH bark, and the lowest were 201.08 g/kg from HH branches. The highest emissions of HC were 532.90 g/kg from BH branches, and the lowest were 75.33 g/kg from BH bark. The highest emissions of $CO_2$ were 1638.28 g/kg from SY branches, and the lowest were 942.49 g/kg from HH trunks. The highest emissions of $NO_x$ were 6.65 g/kg from LYS bark, and the lowest were 0.20 g/kg from HH bark. The highest emissions of $PM_{2.5}$ were 15.61 g/kg from MGL bark, and the lowest were 0.40 g/kg from BH bark, indicating that smoldering promotes $PM_{2.5}$ emissions. Under flaming combustion, the highest emissions of CO were 411.39 g/kg from LYS bark, and the lowest were 149.45 g/kg from BH trunks. The highest emissions of HC were 187.82 g/kg from LYS bark, and the lowest were 34.91 g/kg from BH trunks. The highest emissions of $CO_2$ were 1696.38 g/kg from BH bark, and the lowest were 910.44 g/kg from LYS bark. The highest emissions of $NO_x$ were 8.53 g/kg from MGL trunks, and the lowest were 0.11 g/kg from HH trunks. The highest emissions of $PM_{2.5}$ were 4.57 g/kg from SY branches, and the lowest were 0.29 g/kg from LYS branches. Zhu et al. [43] conducted tests under different experimental conditions, and found that the emissions of $PM_{2.5}$ were 7.2–39.0 g/kg when flaming and 67.6–104.6 g/kg when smoldering, both of which exceeded our corresponding results for $PM_{2.5}$ of 12.48–18.07 g/kg and 14.31–48.40 g/kg when flaming and smoldering, respectively.

This discrepancy might be caused by the experimental devices used and the degree of combustion. The discharge device used in that study was a small-volume chamber, with limited oxygen content and insufficient combustion, causing higher emissions of $PM_{2.5}$. The work by Aurell et al. [42] showed that under the same combustion states, there were significant differences between the emission factors of air sampling and ground sampling. Vicente et al. [15] used operation variables to show that different ignition techniques led to different effects on combustion efficiency; the top-down ignition method had high combustion efficiency and increased the total amount of carbon particles emitted, while the bottom-up ignition method resulted in incomplete combustion of the test material.

Robertson et al. [44] pointed out that there is a positive relationship between temperature and $PM_{2.5}$ emissions, but that $PM_{2.5}$ emissions in winter exceed those in summer. However, in the transition stage from winter to summer, temperature had no significant effect on its emission factors. Therefore, the environmental temperature of sampling may cause a large deviation of emission factors of the litter layer. Compared with the semi-humus layer and the humus layer, we found that the $PM_{2.5}$ emissions of litter were higher, and those from HH were the most significantly pronounced when flaming. In the process of litter combustion, the temperature rises rapidly. Generally, the highest temperature occurs at one-quarter to one-third of the way through the combustion period time, and then the temperature slowly drops [45]. Therefore, the change in combustion temperature may have a significant impact on the gas emissions. When flaming, $CO_2$ emissions from MGL, BH, and LYS were significantly higher than those when smoldering, while $CO_2$ emissions from HH humus when flaming were 2.005 times higher than when smoldering. These differences might be caused by the physiochemical properties of the semi-humus and humus layers, which are strongly affected by microbial decomposition. Many previous studies did not distinguish between different combustion states, and the ranges of $PM_{2.5}$ emission factors obtained from forest combustion for this approach were 5.4–7.2 and 2.32–6.41 $g \cdot kg^{-1}$, respectively [46]. Due to the differences in fuel properties, combustion states, and test devices, there are significant differences between our results and those of other related studies. We found that for smoldering, the highest emissions of $PM_{2.5}$ were from MGL

bark, while the lowest were from BH bark. For flaming, the highest emissions of $PM_{2.5}$ were released from SY branches, and the lowest were from LYS branches. These findings are in contrast to the results of Guo et al. [31] obtained under the same combustion states, where $PM_{2.5}$ released by coniferous species surpassed that of broadleaved species. This may be because of the high combustion efficiency of MGL. Although LYS is a coniferous species, the fire resistance and fire resilience of LYS are higher than those of MGL; it may also be related to the physiochemical properties of the fuel, which need further research and discussion.

In our results, the emissions of $PM_{2.5}$ from MGL bark and SY branches were the most significant, and CO and $NO_x$ emissions from LYS bark were significantly higher than those from other tree species. These species-based differences may be due to the fact that the specific extinction area (SEA) of LYS bark and branches, as well as their total smoke release (TSR), are lower than those of other coniferous tree species [47]. Since the forest fuel sources sampled in this study all grow in the Daxing'an Mountains of Inner Mongolia, our results sufficiently reflect the regional characteristics of fire ecology and behavior.

## 5. Conclusions

A self-designed biomass combustion device (Figure 1) [48,49] was used for an indoor burning experiment. Three main conclusions were found from our investigations:

(1) The overall difference in pollutant emissions released by the combustion of trees under different combustion states ordered from greatest to least was as follows: trunk > branch > bark. The main emission factors were CO, HC, and $CO_2$, whereas $NO_x$ and $PM_{2.5}$ were relatively minor. There were significant differences in emission factors of BH between the two combustion states, while there were significant differences in the emissions of CO, $CO_2$, and $NO_x$ from LYS.

(2) Of the five emission factors, the emission characteristics of $PM_{2.5}$ were the most different, being significantly higher for each tree species under smoldering as compared to flaming. The mass of other emission factors of different tree species increased and decreased in the two combustion states. The total mass of pollutants emitted from the tree species' combustion, ranked from high to low, was as follows: BH > HH > SY > LYS > MGL. Different combustion states had a particularly important effect on the HC released from BH branches.

(3) The emissions of CO, HC, and $PM_{2.5}$ from trees, shrubs, and surface dead fuel under smoldering combustion were significantly higher than those under flaming combustion, while for $CO_2$ and $NO_x$, the opposite was observed. CO emissions were mainly concentrated in the tree bark and the humus layer of surface dead fuel.

Using pollutant emission factor data obtained from our experiments, the emission characteristics and ratio differences of pollutant emission factors from the combustion of different trees, shrubs, and surface dead fuel (and their parts) under two combustion states were comprehensively analyzed. Our results provide both data support and a theoretical basis for regional ecological environment assessments. The indoor combustion experiment showed that pollutants released from combustion are mainly composed of $CO_2$, HC, and CO, but other gases—such as $NO_2$, $NH_3$, and $SO_2$, which are released from actual forest fires—could also have impacts on human health and the ecological environment. Therefore, the emissions of other gases and pollutants should be rigorously investigated in future research.

**Author Contributions:** H.Z. and Q.Z. (Qiuliang Zhang): Conceptualization, Methodology; H.L., X.L., Y.M. and Q.Z. (Qing Zhou): Data Curation, Formal Analysis, Writing—Original Draft Preparation; H.Z.: Investigation, Resources; R.S. and Q.Z. (Qiuliang Zhang): Supervision; R.S. and Q.Z. (Qiuliang Zhang): Writing—Review and Editing. All authors have read and agreed to the published version of the manuscript.

**Funding:** This study was funded by the China Postdoctoral Science Foundation (grant no. 2019M65-3807XB), the National Key Research and Development Project of China (grant no. 2017YFC0504003), and the Inner Mongolia Autonomous Region Science and Technology Plan Project (grant no. 2020GG0067).

**Data Availability Statement:** The data presented in this study are available on request from the corresponding author.

**Conflicts of Interest:** The authors declare that they have no competing financial interest or personal relationships that could have appeared to influence the work reported in this manuscript.

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
