# Peer review of "Emissions Released by Forest Fuel in the Daxing’an Mountains, China"

_forests, doi:10.3390/f13081220_

Round 1

Reviewer 1 Report

The manuscript entitled “Emissions of flue gas and particulate matter from forest fuel in Daxing’an Mountains, China” presents the application of a self-designed biomass combustion system to simulate the combustion of different parts of five tree species and three layers of surface dead fuel of three shrub species.

The topic can be interesting for experts and researchers of vegetation fire emission.

Please find general comments that could help the Authors in improving the manuscript.

The introduction is verbose and somehow redundant (e.g L19-24) I would suggest restructuring it as follows: 1) fire role in the ecosystems 2) fire emission and impacts 3) difference in smoldering and flaming 4) factors influencing smoldering and flaming.

The research question is too wide, in my opinion. “Here self-designed biomass combustion equipment is used to simulate the combustion of different trees, shrubs and surface dead fuel in the Daxing’an Mountains, Inner Mongolia, and the emission characteristics of forest fire emissions of CO, HC, CO2, NOx and PM2.5 and the correlation ratios between various emission factors were analyzed under the two combustion states of smoldering and flaming to provide a scientific basis for evaluating the impact of forest fires on the global climate system and regional air quality.”

Indeed, this type of study is necessary to obtain accurate emission factors for several species, and these values can be used conveniently to estimate fire emission at the local level, but there are different degrees of separation between this aspect and the “evaluation of the impact of forest fires on the global climate system and regional air quality.”

The figures are not clear. According to the text, they should represent a correlation (e.g. “The emissions of CO released by combustion were inconsistent, showing a significant negative correlation with HH branches and BH bark of (Fig. 3a)), although the x-axis title is “mean effect size”. Would you please clarify? Also in the caption is written “If the error line does not intersect with the zero line, it indicates that there is a significant difference between treatment and control” but t is not clear which is the treatment and which is the control. Later on, in the text, you mention “significant differences between the two combustion states. Could you please clarify?

Section 3.2 – Here it is not clear why a linear regression between EF and combustion states was performed. Which is the goal? Also, it is not clear the R value. What do you mean by maximum value? Of R2? Also, you have six groups of material per combustion phase, is this enough to perform regression analysis?

Conclusion section. Points 1 and 2 are more suitable for the discussion section. In the conclusion is not recommendable to summarize the results. The authors should try to extract the main messages from their experiment

Some other considerations are provided to the authors in order to improve the manuscript for further submission.

· Flue gas is normally used to identify the gas exiting the atmosphere via a flue, thus a pipe or channel for conveying exhaust gases from a fireplace, oven, furnace, steam generator or power plant. In the case of forest fires, it is more convenient to use “emission”. For example here “The flue gas released by the combustion of forest fuel”, it would be better to write “the emissions released etc, etc”.

·     Instead of Mhm2 or hm2 please use m2, ha, or km2

·  In the results section, acronyms never presented before are reported, such as HH or BH etc. etc. Please specify what you are referring to since the beginning.

Author Response

Point 1: The introduction is verbose and somehow redundant (e.g L19-24) I would suggest restructuring it as follows: 1) fire role in the ecosystems 2) fire emission and impacts 3) difference in smoldering and flaming 4) factors influencing smoldering and flaming.

Response 1: We are very grateful for your recognition of the research contents and your valuable suggestions. We have reorganized the preamble according to the four aspects you suggested, and the description has become clearer after the revision.

Point 2: The research question is too wide, in my opinion. “Here self-designed biomass combustion equipment is used to simulate the combustion of different trees, shrubs and surface dead fuel in the Daxing’an Mountains, Inner Mongolia, and the emission characteristics of forest fire emissions of CO, HC, CO2, NOx and PM2.5 and the correlation ratios between various emission factors were analyzed under the two combustion states of smoldering and flaming to provide a scientific basis for evaluating the impact of forest fires on the global climate system and regional air quality.”Indeed, this type of study is necessary to obtain accurate emission factors for several species, and these values can be used conveniently to estimate fire emission at the local level, but there are different degrees of separation between this aspect and the “evaluation of the impact of forest fires on the global climate system and regional air quality.”

Response 2: Thanks for pointing this out,we have deleted "evaluation of the impact of forest fires on the global climate system and regional air quality" and replaced it with a more convenient estimate of local fire emissions (lines 128-133)

Point 3: The figures are not clear. According to the text, they should represent a correlation (e.g. “The emissions of CO released by combustion were inconsistent, showing a significant negative correlation with HH branches and BH bark of (Fig. 3a)), although the x-axis title is “mean effect size”. Would you please clarify? Also in the caption is written “If the error line does not intersect with the zero line, it indicates that there is a significant difference between treatment and control” but t is not clear which is the treatment and which is the control. Later on, in the text, you mention “significant differences between the two combustion states. Could you please clarify?

Response 3: After some reflection we removed the graph made by the meta-analysis and replaced it with a histogram drawn using one-way ANOVA, which made the results clearer and easier to understand(Figure 3-Emissions of released by five trees in different combustion states, Figure 4-Emissions of released by three shrub species under different combustion states,Figure5-Emissions of released by different layers of five tree species under different combustion states.)

Point 4: Here it is not clear why a linear regression between EF and combustion states was performed. Which is the goal? Also, it is not clear the R value. What do you mean by maximum value? Of R2? Also, you have six groups of material per combustion phase, is this enough to perform regression analysis?

Response 4: Thank you for your comments. The purpose of the article is to explore the effects of different tree species, different humus levels, different stand types, and tree parts on combustion measurements such as CO in the shaded and open combustion states. Since tree species, humus levels, stand types, and combustion parts are definite class variables, we found that the idea of using linear regression is not in line with scientific data analysis methods and cannot obtain scientific results, which is not consistent with the actual purpose. We therefore used one-way ANOVA to investigate the effect of different tree species and humus levels on burning indexes.

Point 5: Conclusion section. Points 1 and 2 are more suitable for the discussion section. In the conclusion is not recommendable to summarize the results. The authors should try to extract the main messages from their experiment

Response 5: We quite agree with your advice. We have put the first and second points of the conclusion into the discussion and have reworked the discussion to avoid the issues you raised.

Point 6: In the case of forest fires, it is more convenient to use “emission”. For example here “The flue gas released by the combustion of forest fuel”, it would be better to write “the emissions released etc, etc”.

Response 6: Thank you for pointing this out. We have changed “flue gas” in the full text to “emissions released”.

Point7: Instead of Mhm2 or hm2 please use m2, ha, or km2

Response 7: Thank you for pointing this out. All the places where Mhm2 and hm2 appear in the text have been replaced with ha and km2 (Line 77, Line 117, Line 147).

Point 8: In the results section, acronyms never presented before are reported, such as HH or BH etc. etc. Please specify what you are referring to since the beginning.

Response 8: Thank you for pointing this out. We added letter combinations to 2.2 First occurrence of different tree species, different letter combinations represent different tree species, e.g. Quercus mongolica (MGL), MGL stands for Quercus mongolica.(lines 153-157)

Reviewer 2 Report

The subject matter covered by the authors is interesting. I would suggest changing the keywords to e.g.: biomass combustion, wood combustion, controlled combustion, incomplete combustion, 

What is missing from the introduction, in my view, is information on the adaptation of certain forest complexes to the occurrence of cyclical fires, e.g. sequoias, which spread their seeds by this mechanism, savannah forests, etc. In addition, there is no information about fires such as the peat fire in northern Yakutia, which continues to generate greenhouse gases in the atmosphere, and there is no idea how to put them out. 

The second element worth mentioning in the introduction is the growing use of woody biomass in the energy sector. When burned in specialised cookers, such fuel does not contribute to the emission of significant levels of harmful compounds. This situation changes when traditional cookers and furnaces are used for heating, e.g. in homes.   

The methodology outlines the sampling method for incineration. The method of drying the material before combustion is also presented here. In my opinion, information on the initial moisture content of the material tested should be included here. The very process of drying lignocellulosic biomass to a constant mass allows to reduce the formation of some of the harmful organic compounds that will be formed as a result of the combustion of the forest complex. This process will certainly reduce the formation of harmful compounds and reduce the content of e.g. carbon monoxide in the results.  

Figures 3,4,5 are difficult to read. I would suggest changing them to a column chart or tables for example. 

The conclusions are the most important truths revealed by the experiment. They should not reiterate the purpose of the research, the methodology, the description of the research apparatus. In my opinion, there should be information whether the apparatus allows carrying out the planned experiments. Then, which type of combustion is the most beneficial and which generates the most toxic compounds. 

Examples of articles that may be useful.

https://doi.org/10.1016/B978-0-444-59510-2.00001-X

DOI:10.3390/en14102968

DOI:10.1016/S1474-8177(08)00004-1

Author Response

Point 1: The subject matter covered by the authors is interesting. I would suggest changing the keywords to e.g.: "biomass combustion, wood combustion, controlled combustion, incomplete combustion".  

Response 1: Many thanks for the constructive feedback on our manuscript. We have changed the keywords .

Point 2: What is missing from the introduction, in my view, is information on the adaptation of certain forest complexes to the occurrence of cyclical fires, e.g. sequoias, which spread their seeds by this mechanism, savannah forests, etc. In addition, there is no information about fires such as the peat fire in northern Yakutia, which continues to generate greenhouse gases in the atmosphere, and there is no idea how to put them out. 

Response 2: Thank you for the suggestion. The relevant example has been added and described as "Wildfires and prescribed fires (preceded by harvest or not) can serve to promote giant sequoia regeneration, providing that fire intensity is sufficient to create canopy gaps, increase understory light, and remove surface litter.[1] In boreal North America, black spruce shapes forest flammability and depends on fire for regeneration[2]. Transition of forests to savannah suggest that fire disturbance can be a major driver of biome change[3]."(line 53-60)  

"he presence of open flame—i.e., flaming combustion—has a significant impact on the chemical composition of emissions and plume dynamics[4]. Smoldering combustion is the driver of wildfires in peat lands, such as those that cause episodes of haze in Southeast Asia, North America, and Northeast Europe[5]."(line 64-68)

References:

  1. Meyer, M.D.; Safford, H.D. GIANT SEQUOIA REGENERATION IN GROVES EXPOSED TO WILDFIRE AND RETENTION HARVEST. Fire Ecology 2011, 7, 2-16, doi:10.4996/fireecology.0702002.
  2. Baltzer, J.L.; Day, N.J.; Walker, X.J.; Greene, D.; Mack, M.C.; Alexander, H.D.; Arseneault, D.; Barnes, J.; Bergeron, Y.; Boucher, Y.; et al. Increasing fire and the decline of fire adapted black spruce in the boreal forest. 2021, 118, e2024872118, doi:doi:10.1073/pnas.2024872118.
  3. Aleman, J.C.; Blarquez, O.; Elenga, H.; Paillard, J.; Kimpuni, V.; Itoua, G.; Issele, G.; Staver, A.C. Palaeo-trajectories of forest savannization in the southern Congo. Biology Letters 2019, 15, doi:10.1098/rsbl.2019.0284.
  4. 4.Urbanski, S.P.; Wei, M.H.; Baker, S.J.D.i.E.S. Chapter 4 Chemical Composition of Wildland Fire Emissions. 2008, 8, 79-107.DOI:10.1016/S1474-8177(08)00004-1

5.Rein, G. Smoldering-Peat Megafires: The Largest Fires on Earth; Coal and Peat Fires: a Global Perspective: 2015.https://doi.org/10.1016/B978-0-444-59510-2.00001-X

Point 3: The second element worth mentioning in the introduction is the growing use of woody biomass in the energy sector. When burned in specialised cookers, such fuel does not contribute to the emission of significant levels of harmful compounds. This situation changes when traditional cookers and furnaces are used for heating, e.g. in homes.

Response 3: Thank you for your valuable comments. We have included the growing use of woody biomass in the energy sector in the foreword (Roman, K.2021).(Line 123-127)

References:

Roman, K.; Roman, M.; Szadkowska, D.; Szadkowski, J.; Grzegorzewska, E. Evaluation of Physical and Chemical Parameters According to Energetic Willow (Salix viminalis L.) Cultivation. Energies 2021, 14, doi:10.3390/en14102968.  

Point 4: The methodology outlines the sampling method for incineration. The method of drying the material before combustion is also presented here. In my opinion, information on the initial moisture content of the material tested should be included here. The very process of drying lignocellulosic biomass to a constant mass allows to reduce the formation of some of the harmful organic compounds that will be formed as a result of the combustion of the forest complex. This process will certainly reduce the formation of harmful compounds and reduce the content of e.g. carbon monoxide in the results.

Response 4: We understand the reviewer’s concern. Thank you for your constructive comments and suggestions. We previously ignored the issue of biomass fuel moisture content, but we actually correlated the combustibles prior to the experiment because the study was not concerned with the effect of biomass moisture content on pollutant emissions. All samples were naturally air dried to a constant weight over three days.(line161-164)

Point 5: Figures 3,4,5 are difficult to read. I would suggest changing them to a column chart or tables for example. 

Response 5: Thank you for your advice. We have modified the original chart to an easy-to-read bar chart. 

Point 6: The conclusions are the most important truths revealed by the experiment. They should not reiterate the purpose of the research, the methodology, the description of the research apparatus. In my opinion, there should be information whether the apparatus allows carrying out the planned experiments. Then, which type of combustion is the most beneficial and which generates the most toxic compounds.  

Response 6: Thank you for your valuable advice. We have made additions at the conclusion based on your suggestions. This approach has also been adopted by (Guo et al.,2017).

DOI:10.3390/en14102968

References:Guo, F.T., Jin, Q.F., Yang, X.J., Liu, A.Q., 2017. An Air Compression System that Simulates the Burning of Wild Biomass; China. 2016211196373 [P].
